# Enhanced optical conductivity and many-body effects in strongly-driven photo-excited semi-metallic graphite

T. P. H. Sidiropoulos [1,2] ✉, N. Di Palo[1], D. E. Rivas [1], A. Summers[1], S. Severino[1], M. Reduzzi[1] & J. Biegert [1,3] ✉

The excitation of quasi-particles near the extrema of the electronic band structure is a gateway to electronic phase transitions in condensed matter. In a many-body system, quasi-particle dynamics are strongly influenced by the electronic single-particle structure and have been extensively studied in the weak optical excitation regime. Yet, under strong optical excitation, where light fields coherently drive carriers, the dynamics of many-body interactions that can lead to new quantum phases remain largely unresolved. Here, we induce such a highly non-equilibrium many-body state through strong optical excitation of charge carriers near the van Hove singularity in graphite. We investigate the system's evolution into a strongly-driven photo-excited state with attosecond soft X-ray core-level spectroscopy. We find an enhancement of the optical conductivity of nearly ten times the quantum conductivity and pinpoint it to carrier excitations in flat bands. This interaction regime is robust against carrier-carrier interaction with coherent optical phonons acting as an attractive force reminiscent of superconductivity. The strongly-driven non-equilibrium state is markedly different from the single-particle structure and macroscopic conductivity and is a consequence of the non-adiabatic many-body state.

Optically-induced electronic phase transitions in strongly-driven condensed matter systems manifest from an out-of-equilibrium state, such as the optical excitation of phonons and their nonlinear coupling to electronic states[1–5]. However, a very high density of states is required for a phase transition to occur at room temperature. A gateway to induce such new phases are high-momentum electronic states at the edge of the Brillouin zone, where the single-particle electronic band structure exhibits extrema such as flat electronic bands[6,7]. For electronic excitations with energies near a van Hove singularity (VHS), correlated electronic states with properties similar to superconductivity[8,9] or magnetism[10,11] have been recently observed in graphene. Electronic phase transitions in bulk graphite have been reported for intercalated compounds with a Fermi level near the VHS[12,13]. Some works even reported possible phase transitions in highly pyrolytic graphite (HOPG) at elevated temperatures; however, the exact mechanisms remain debated[14–17].

HOPG with an AB (Bernal) stacking of van-der-Waals bound layers is interesting for such investigations as the material shares similar properties with bilayer-graphene; see Fig. 1a. In both systems, inter-layer coupling of electronic states in neighboring layers lifts the degeneracy of the bands at the Dirac point, which leads to split-off bands with a band-gap of less than 60 meV at the K-point[18,19]. In contrast, the bands remain linear around the H-point, i.e., charge carriers behave like massless Dirac-particles[20]. The split-off bands at the K-point are connected to the H-point by a near dispersion-less band, removing the electronic system's two-dimensional (2D) confinement;

[1]ICFO - Institut de Ciencies Fotoniques, The Barcelona Institute of Science and Technology, 08860 Barcelona, Spain. [2]Max-Born-Institut für Nichtlineare Optik und Kurzzeitspektroskopie, 12489 Berlin, Germany. [3]ICREA - Institució Catalana de Recerca i Estudis Avançats, Barcelona, Spain. ✉ e-mail: sidiropo@mbi-berlin.de; jens.biegert@icfo.eu

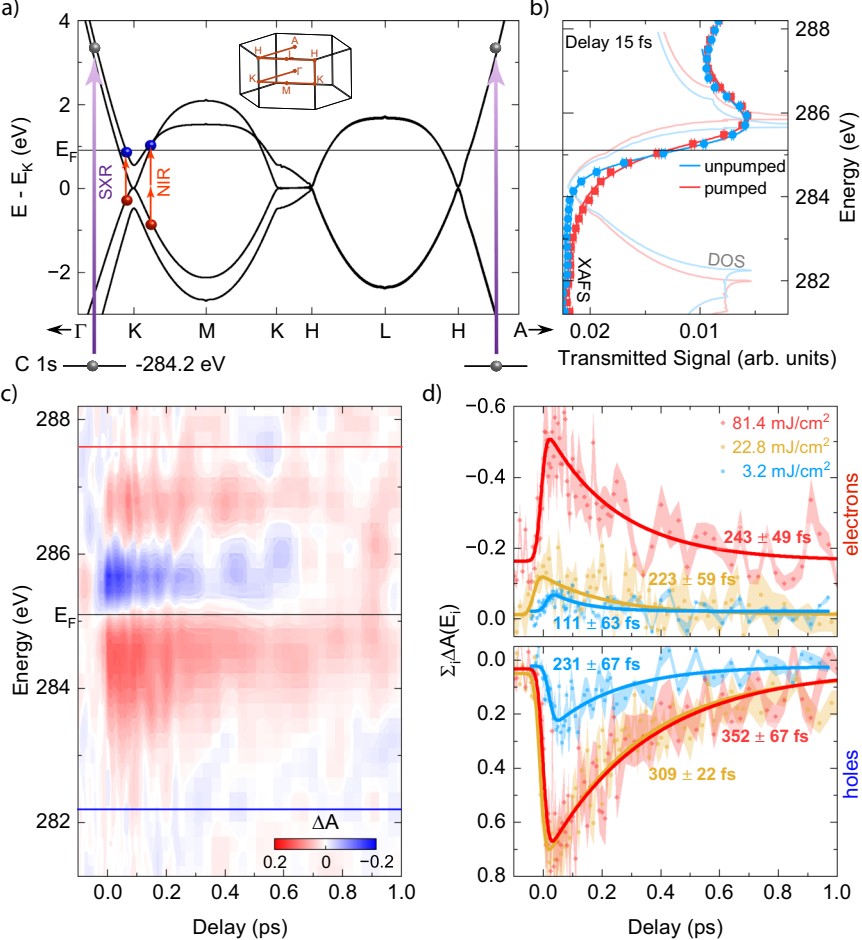

**Fig. 1 | Signatures of the electronic structure in time-resolved XAFS. a** Single-particle electronic band structure of AB stacked highly pyrolytic graphite (HOPG) calculated with Wien2K[65]. The energy is referenced to the K-point. The inset shows graphite's hexagonal unit cell with the path for which the band structure is presented. The near-infrared beam excites carriers around the Fermi energy ($E_F$), and the soft X-ray beam probes changes in the carrier occupation by promoting 1 s core-electrons that are bound by 284.2 eV into free electronic states around $E_F$ (horizontal line). **b** Retrieved density of states from the fit for the unpumped case (light blue line) and pumped case, 15 fs after near-infrared excitation with 81.4 ± 5 mJ/cm² (light red line). Symbols are the measured XAFS with (red squares) and without

(blue circles) near-infrared excitation and the corresponding fits (solid lines). **c** $\Delta A$ for excitation with 81.4 ± 5 mJ/cm² (high fluence case). The black line indicates the position of the static Fermi energy at 285.1 eV, as retrieved from the fit. The red (blue) line at 287.6 eV (282.2 eV) marks the upper (lower) energy limit for the lineouts in (**d**). **d** $\Sigma_i \Delta A(E_i)$ above (negative) and below (positive) the static $E_F$ from (**c**) for the three measured fluences and the corresponding decay times from exponential fits (solid lines). The error bars are the relative error of the mean of $\Delta A(E_i)$ in the corresponding electron and hole energy range. Source data are provided as a Source Data file.

thus, through interlayer coupling the carriers occupy 3D Fermi surfaces[19,21–23]. Away from the K(H)-point towards the M(L)-point, graphite's single-particle structure leads to flat bands, which manifest as the VHS with a very high density of states (DOS). These bands may be accessed by carrier doping or by controlling the twist angle between graphene layers−both result in moving the Fermi level near the VHS. The Fermi surface becomes two-dimensional, and correlated electron effects are observed[6,8,10,11].

Here, we will identify such change of the electronic phase in a light-induced out-of-equilibrium state[8,9,13–15] by time-resolving changes in the dimensionality of the Fermi surface of the many-body quasi-particle system. While photoinduced phase transitions are easily identified through the optical conductivity in spectroscopic measurements of the sample reflectivity[2–4], such measurement can provide only indirect information to infer a carrier distribution[4,24,25]. In contrast, photoemission spectroscopies[26–28] are ideal for directly measuring the single-particle structure from surfaces. However, the inelastic scattering of photoexcited electrons for a bulk system makes such measurement challenging. In contrast, we time-resolve the entire out-of-equilibrium state of the system and study the energy-dependent

relaxation dynamics of quasi-particles for different photo-doping with attosecond soft X-ray absorption spectroscopy (XAFS).

Considering that the self-energy, $\Sigma(k, \omega)$ describes the polarization of the electronic system and quasi-particle excitations such as carrier-carrier or carrier-phonon interactions[29–32], the direct relation between the self-energy and the inverse of the single-particle scattering time connects the measured single-particle spectrum to the quasi-particle dynamics,

$$\frac{1}{\tau_{n,k}} = -\frac{2\Im\Sigma(E_{n,k})}{\hbar} \tag{1}$$

where the spectral function is related to the self-energy through $A(\mathbf{k}, E) = -2\Im\Sigma(E_{n,k})/[(E - E_{n,k} - \Re\Sigma(E_{n,k}))^2 + \Im\Sigma(E_{n,k})^2]$. Consequently, the energy scaling of the single-particle scattering time $\tau_{n,k}(E)$ allows the identification of the quasi-particle system and its electronic phase. For instance, a 3-dimensional (3D) Fermi-liquid with carrier-carrier scattering is characterized by a rate $\tau_{n,k}(E)$, which scales quadratically with energy $1/\tau \sim (E - E_F)^2$ [32]. Since the exact scaling depends on the dimensionality of the studied electronic system, this

allows us to denote the type of quasi-particle interactions and the electronic single-particle band structure[31,33,34]. For example, low photo-doping leads to a deviation from the simple $\sim (E - E_F)^2$ scaling in graphite[18,24,35] and graphene[26], with clear signatures of carrier-carrier, carrier-phonon, and plasmon excitations, as well as the renormalization of the band structure. These are clear signatures of many-body effects that can lead to entirely new quantum phases under strong optical excitation[36–38]. In the following, we will use attosecond XAFS to resolve the influence of the single-particle structure on the quasi-particle dynamics of a strongly-driven photo-excited state. The relation between the quantum and optical conductivity allows the identification of the phase-transition when carriers occupy the extrema of the single-particle band structure.

Briefly, XAFS provides access to a wide range of electronic states[39] by probing the unoccupied density of states. The method employs transitions from atomic core states to free electronic states near the Fermi edge, giving information about the single-particle electronic structure and the lattice configuration[40]. To avoid multiple absorption pathways from energetically close atomic core states, measuring the absorption, $A(E)$, from the 1-s (K-edge) core state is crucial, as it gives an unclouded view of the unoccupied electronic states described by the product of Fermi-Dirac distribution with the single-particle density of states, $A(E) \approx [1 - f(E)]*D(E)$[41,42]. Photo-excitation of carriers into free electronic states then modifies the momentum averaged X-ray absorption spectrum, $A_p(E)$[33]. We probe the time-dependent evolution of the carrier occupation as $\Delta A(E, t) = A_p(E, t) - A(E, t)$. The single-particle scattering time $\tau_{n,\mathbf{k}}$ is related to the measured many-body relaxation time, $\tau$, of the carrier occupation through the Boltzmann transport equation[31,43,44]. And similar to the self-energy, quasi-particle excitations are described through the memory function $M$[5,45–47] with

$$\frac{1}{\tau(E)} = \frac{\Im M(E)}{\hbar} \qquad (2)$$

This equation resembles Eq. (1), and indeed such a relation between $\Im M(E)$ and $\Im \Sigma(E_{n,\mathbf{k}})$ has been predicted[46]. While the exact relation between the spectral function and the memory function is unknown, the many-body relaxation rate is obtained from the decay of the carrier occupation, $\partial_t \Delta A(E, t)$. The complex memory function, $M(E) = \frac{E}{\hbar}\lambda(E) + \frac{i}{\tau(E)}$ is then linked to the optical conductivity:

$$\sigma(E) = \frac{i\varepsilon_0 \omega_p^2}{M(E) + E/\hbar} = \frac{\varepsilon_0 \omega_p^2}{\frac{1}{\tau(E)} - iE/\hbar(1+\lambda(E))} \qquad (3)$$

Where $1 + \lambda(E) = m^*/m_e$ is the mass enhancement that arises through many-body interactions and is related to the thermalization rate through a Kramers-Kronig transformation[47]. Thus, time-resolved XAFS relates the time-dependent carrier occupation $\Delta A(E, t)$ to the optical conductivity $\sigma(\omega)$. The transient transmitted soft X-ray spectrum through optically excited graphite gives access to the complex optical conductivity, which describes quasi-particle excitations in a many-body spectrum.

## Results and discussion

Figure 1b shows the results of the near-edge soft X-ray absorption measurement of 95-nm thick graphite with a 165-as soft X-ray pulse. We interrogate the sample with a linearly-polarized soft X-ray pulse under 40° incident angle to access the $\pi^*$-bands in graphite along K-M and A-H directions. The rise in absorption at 284.2 eV is near the K(H)-point, and the peak near 285.7 eV is the signature of the flat bands near the M(L)-point. For reference, we superimposed the calculated density of states. We infer the sample's intrinsic n-doping and Fermi energy at 285.1 eV from a fit of the absorption spectrum to a room-temperature Fermi-Dirac distribution multiplied by the density of states; see Supplementary Eq. 1. To extract the dynamic evolution of carriers and their

many-body physics, we leverage the direct mapping between measured core-level soft X-ray spectrum and unoccupied density of states with the broadband soft X-ray pulse after optical excitation with a two-cycle (11.3-fs) near-infrared pump pulse with a central energy of 0.7 eV. Figure 1c shows the change in absorbance, $\Delta A$, for delays of up to 1 ps between the soft X-ray probe and pump pulse, polarized in the hexagonal graphite lattice's basal plane (G-K-M).

The measurement shows that although the pump pulse can only excite carriers along the K-G direction [see Supplementary Note 2], multi-photon absorption[48,49], phonon-assisted absorption[50], and carrier multiplication[38] lead to the ultrafast occupation of electronic states near the M-point. To study the many-body response properties of strongly-driven photo-excited graphite, we investigate the system under extreme optical excitation with absorbed fluences up to 81.4 $\pm$ 5 mJ/cm² . Such extreme photo-doping with $n_{opt} \approx 5 \times 10^{22}$ electrons/cm³ [38] significantly exceeds the number of free electronic states (~$10^{19}$ cm⁻³)[19], thus effectively altering the single-particle electronic structure in highly photo-excited graphite with non-equilibrium carrier dynamics[27,28]. Figure 1d shows lineouts of the absorption measurement under such conditions, exhibiting asymmetric dynamic evolution of the states above (electrons) and below (holes) the Fermi energy. Note, for an excitation with 81.4 $\pm$ 5 mJ/cm² we have considered the total positive (negative) change in $\Delta A$ for states below (above) the Fermi energy. An exponential fit to the energy-integrated absorption spectrum over the apparent signal, $\Sigma_i \Delta A(E_i, t)$, for three pump fluences of 3.2 $\pm$ 0.2, 22.8 $\pm$ 1.4 and 81.4 $\pm$ 5 mJ/cm² reveals that extreme optical doping increases the relaxation times with electrons losing energy faster than holes. Compared to single-particle scattering times from photoemission measurements on graphite, these relaxation times are by one order of magnitude larger and highlighting XAFS as a probe of graphite's non-equilibrium many-body response[18,30,51]. The initial non-thermal energy distribution of optically excited carriers rapidly thermalizes through carrier-carrier scattering and strong coupling to optical phonons[38]. However, further relaxation of hot carriers slows down due to the small density of states at the K(H)-point, the considerable optical phonon energies, and their slow cooling through phonon-phonon scattering with typical reported carrier relaxation times of 200-250 fs[24,33,35,52,53]. The large intrinsic doping of our sample, however, initially avoids these bottlenecks as the Fermi-energy is above the K-point where the density of states is larger. This is apparent from the short electron (hole) decay time of 111 $\pm$ 63 fs (231 $\pm$ 67 fs) for the low fluence case. Only by increasing the fluence to 22.8 $\pm$ 1.4 mJ/cm² the electron (hole) relaxation time increases to 232 $\pm$ 82 fs (309 $\pm$ 22 fs) as low energetic states become occupied and carrier relaxation becomes limited by the slow cooling of optical phonons. With a further increase in fluence to 81.4 $\pm$ 5 mJ/cm², another bottleneck occurs as carriers occupy flat bands, where carrier relaxation can only occur through transferring a significant crystal momentum by scattering with optical phonons[35].

We now employ the sensitivity of XAFS to graphite's many-body response, together with the ability to separate the carrier dynamics from the renormalization of the single-particle density of states through the fit, to reveal the underlying quasi-particle excitations. Our model of the transient XAFS signal further accounts for carrier-induced changes to the density of states and allows us to separately identify contributions of carrier temperature, Fermi-edge position, and shift of the VHS at each pump-probe delay; further details are provided in the Supplementary Note 4.

A comparison of the transient changes in the density of states between the three measured fluence cases, shown in Fig. 2d and h, directly highlights the carrier density-dependent effects. We note that our time-resolved fit reveals a shift of the density of states to higher energies of 200 meV and contradicts reported photoinduced band-gap renormalization values in graphite[24,25,28,54]. Contrary to measurements that cannot distinguish between different carrier types, we use

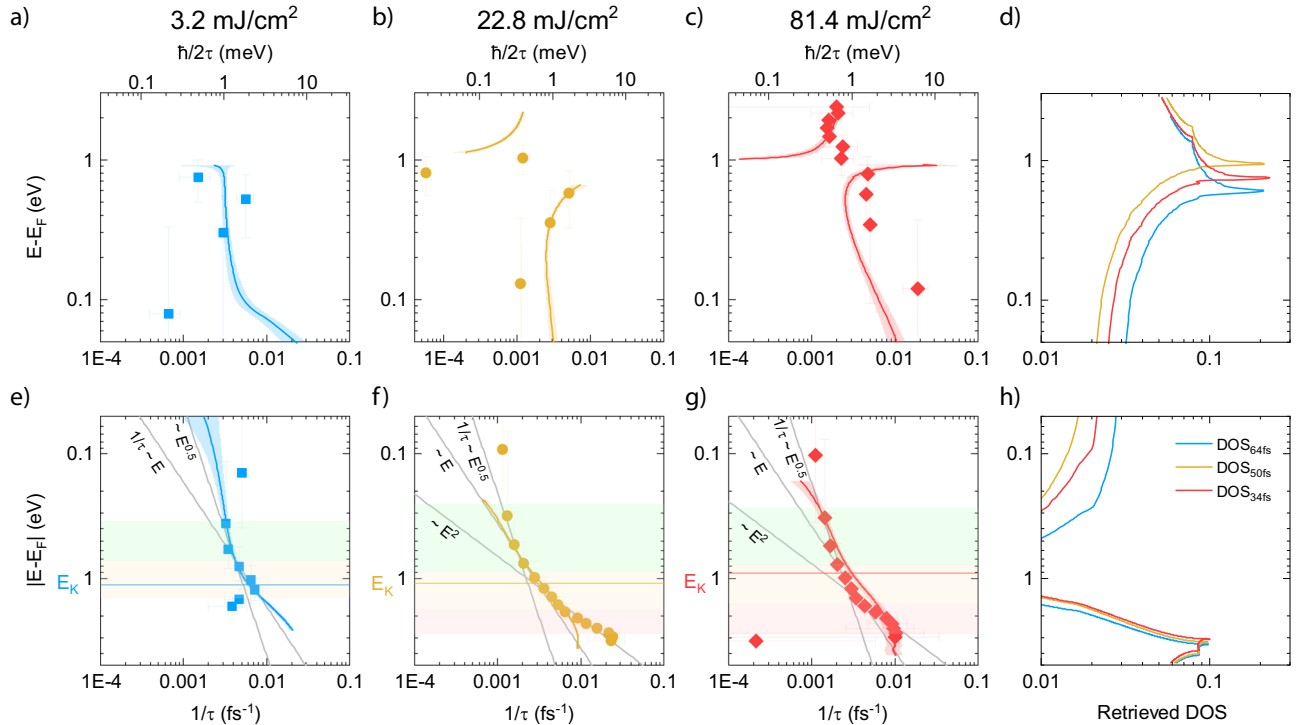

**Fig. 2 | Carrier thermalization rates after optical excitation.** Carrier thermalization rate $1/\tau_{th}$ from an exponential fit to $\Delta A$ for all three measured fluences (symbols) for electrons **a**–**c**, holes **e**–**g**, and from the fitted absorption spectra (solid lines). The energy axis is referenced to the static Fermi energy. The horizontal lines in (**e**–**g**) indicate the position of the K-point. The gray lines indicate the slope of $1/\tau$ in the apparent energy region (colored boxes), highlighting the dimensionality of the carrier system. The spectral resolution determines the error bars along the energy axis and the error in the thermalization rate is derived from the exponential fit to the measured and retrieved $\Delta A$. **d**, **h** The renormalized density of states retrieved from the model at the delay time at which the carrier temperature becomes maximal. Source data are provided as a Source Data file.

this novel capacity to determine the origin of the DOS shift and the underlying influence of the single-particle spectrum. We extract the energy-dependent carrier relaxation $\tau(E) = |\partial_t \Delta A(E,t)|_{max}/e$ from the XAFS measurement and find excellent agreement between $1/\tau(E)$ and the retrieved relaxation rate, shown in Fig. 3. We confirm the relation between $\Im M(E)$ and $\Im \Sigma(E_{n,\mathbf{k}})$ described above, by identifying that the measured carrier relaxation in n-doped graphite follows a similar trend as the self-energy of n-doped graphene. It increases continuously away from the Fermi level, with anisotropic carrier-carrier and carrier-phonon contributions for electrons and holes[34]. For holes ($E < E_F$), the relaxation rate in Fig. 3e, f reveals a continuous increase with energy, but its scaling changes. For low-energy holes $E \leq (E_F)$, $1/\tau$ scales as $\sim (E - E_F)^{1/2}$, it becomes linear for energies near the K-point ($E < E_F$), and it exhibits quadratic scaling close the hole VHS ($E \ll E_F$). Interestingly, the renormalization of the density of states is screened close to $E_K$ and does not influence the hole thermalization rate. In contrast to holes, the relaxation rate for electrons ($E > E_F$) is discontinuous. For fluences of up to 22.8 $\pm$ 1.4 mJ/cm² the relaxation rate increases with electron energy with a sudden decrease for energies above the renormalized VHS, see Fig. 3d. We attribute this behavior to the proximity of the Fermi energy to the flat-bands. Unexpectedly, for the most strongly-driven system, at 81.4 $\pm$ 5 mJ/cm², we observe a sudden stabilization of the electron relaxation rates, independent of energy and with only a minimal decrease above the VHS. Our measurement explains the found behavior of quasi-particle lifetimes and conductivity. For such extreme optical excitation, a Fermi level near the VHS, where the electronic bands flatten, results in drastically increased quasi-particle lifetimes, as carrier-carrier interactions are effectively suppressed. Under this regime, phonons act similarly to an attractive force and become the main relaxation channel for electrons[26,33–35]. This effect is discussed as artificially induced superconductivity in the

context of twistronics[7,55]. In contrast, for holes, the split-off bands near the K-point, and the linear bands between the K-point and the hole VHS, lead to vastly different relaxation rates as carrier-carrier scattering becomes the main relaxation channel[34]. Besides, the sudden decrease in the relaxation rate below the K-point for the lowest photo-doping with 3.2 $\pm$ 0.2 mJ/cm² is due to the plasmon resonance[26]. This peak shifts towards higher energies as the carrier density increases. As the hole occupation increases and reaches the flat bands, carrier-carrier scattering becomes the main relaxation channel and $1/\tau(E)$ resembles the single-particle structure.

Finally, we leverage the similarity between the self-energy and the single particle density of states[43] to assess the general charge carrier dispersion over the apparent energy range and the dimensionality of the charge carrier system from the scaling of the relaxation rates (see Supplementary Table 3). We find that near the Fermi energy, where the split-off bands approximate parabolic behavior, the $E^{1/2}$ scaling of $1/\tau(E)$ indicates a 3D hole system. This pinpoints the 3D Fermi surface to the H-point. We note that this finding agrees with existing magneto-transport and photoemission measurements[20–22]. Further from the Fermi energy, around the K(H)-point, we infer from the linear scaling a change in the dimensionality to a 2D-like carrier system with linear dispersion[33]. However, specifically for energies approaching the VHS, holes behave again like a 3D system, as carriers occupy states beyond the split-off bands, arising through interlay coupling, and where the bands approaching the VHS become linear[56].

In contrast to holes, we attribute the rather peculiar scaling of electrons to carrier occupation near the VHS, where the bands are essentially flat. The electron relaxation rate increases further from the Fermi level for weak optical interaction due to strong electron-electron and electron-phonon scattering[34]. In the regime of a strongly-driven photo-excited state, i.e., for photo-doping with 81 $\pm$ 5 mJ/cm², the

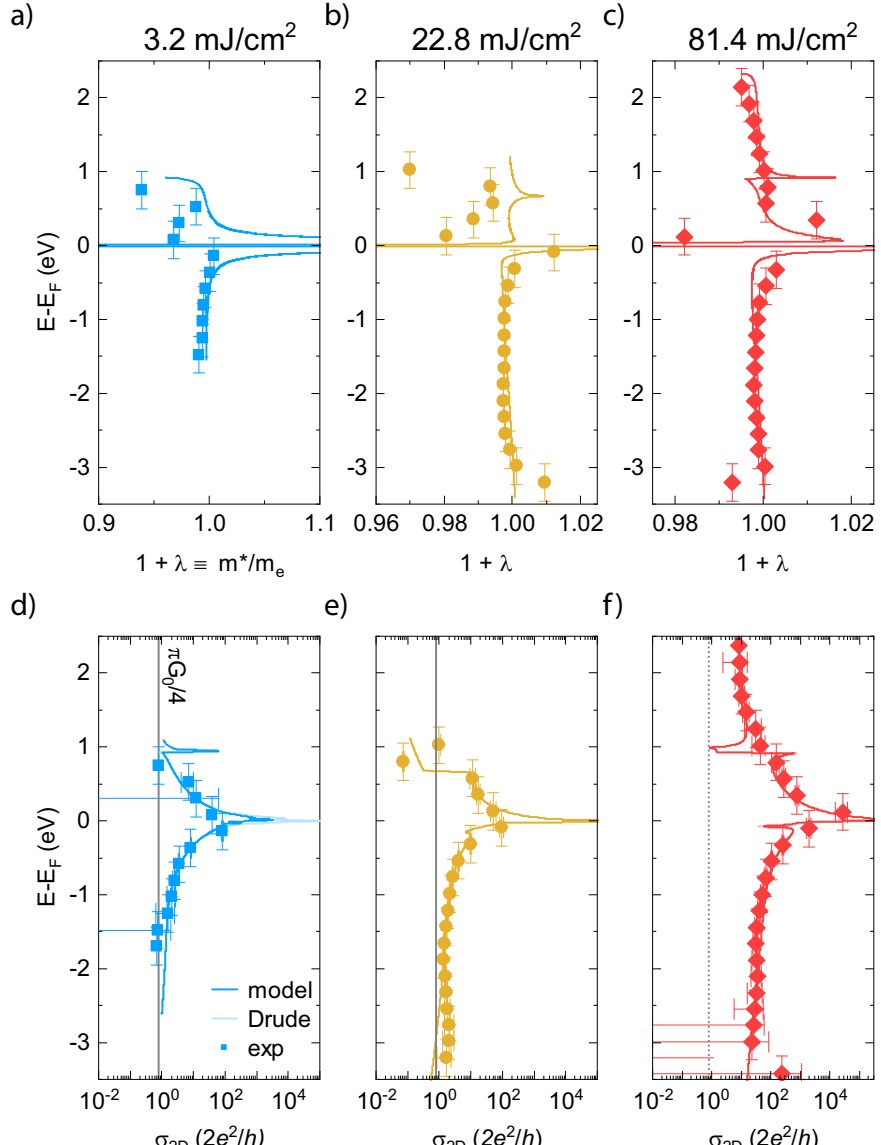

**Fig. 3 | Optical conductivity and mass enhancement. a–c** Mass enhancement factor $\lambda(E)$ and optical conductivity $\sigma(E)$ (**d-f**) retrieved from the relaxation rate, $1/\tau(E)$, using a Kramers-Kronig transformation of the experimental data (symbols) and the model (lines). The vertical line in (**d-f**) marks the universal optical conductivity of a single graphene layer at $\pi G_0/4$. The light line in (**d-f**) is the Drude conductivity. The error bars along the energy axis are determined by the spectral resolution. The error in the optical conductivity is determined by the propagated error of the thermalization rate from Fig. 2. Source data are provided as a Source Data file.

strong many-body interaction dominates and leads to stabilization of $1/\tau$. The stabilization in the strongly-driven photo-excited state is reminiscent of a 2D parabolic electronic system and suggestive of strong suppression of interlayer hopping for electrons above the VHS[6,57]. We explain this behavior with Pauli blocking low energy states and the flattening of bands near the VHS, which suppresses electron-electron interactions due to the requirement for transfer of considerable in-plane momentum by scattering with optical phonons[33–35,38].

We now use the relation between the complex memory function and the macroscopic optical conductivity to identify the quasi-particle interactions in strongly-driven photo-excited graphite. A Kramers-Kronig transformation of the carrier relaxation rate $\Im M = 1/\tau(E)$ (Fig. 3a–c) yields the mass enhancement factor $\Re M = 1 + \lambda(E)$ which is a clear signature of many-body interactions; the effective mass scales as $1 + \lambda = m^*/m_e$. Interestingly, despite the very strong coupling to optical phonons, we observe, independent of photodoping, no significant increase in the effective mass, $m^*$. The exception is energies near the Fermi-energy and the flat bands, for which the electronic mass varies between 0.95 $m_e$ and 1.2 $m_e$. This is the case even for weak excitation with 3.2 ± 0.2 mJ/cm². Fig. 3d–f show the measured mass enhancement factors and optical conductivity in units of the 2D quantum conductivity $G_0 = 2e^2/h$. The figures reveal a Drude-like discontinuity in the optical conductivity close to the Fermi energy, which remains for holes over a broad energy range near $G_0$. Increasing the fluence to 81.4 ± 5 mJ/cm², the conductivity increases by one order of magnitude to $10G_0$. We observe an entirely different behavior for electrons. For fluences of up to 22.8 ± 1.4 mJ/cm², transitions into the flat bands near the VHS are apparent as peaks in the optical conductivity around 1 eV above the Fermi level. But, for the most strongly-driven system, at 81.4 ± 5 mJ/cm², σ follows a Drude-like behavior with values approaching the ones of holes and only a kink near the VHS.

We contrast the observed behavior under weak optical excitation against data in graphene. In graphene, a frequency-independent

optical conductivity of $\pi G_0/4$ arises from massless Dirac carriers near the K-point[54,58–60]; in graphite, however, the linear bands remain degenerate only at the H·point[19,20]. The large n-doping of the graphite sample leads to the occupation of all states along the K-H direction, effectively suppressing the energy dependence of the conductivity[58]. Discontinuities in the optical conductivity of graphene away from the Fermi level arise from electronic excitations into states near the VHS[58–61]. Therefore, in the weak excitation regime, the optical conductivity of graphite highlights the similarity to graphene. Under the most extreme optical excitation, however, carrier transport and conductivity stabilize independently of the carrier concentration. This interaction regime is robust against carrier-carrier interaction with coherent optical phonons acting as an attractive force reminiscent of superconductivity. The strongly-driven non-equilibrium state is markedly independent of the single-particle structure and is a consequence of the non-adiabatic many-body state.

This work investigates the non-equilibrium many-body dynamics of semi-metallic graphite under extreme photo-doping with attosecond soft X-ray core-level spectroscopy. We find a surprising enhancement by one order of magnitude of the optical conductivity and pinpoint it to excitation in flat bands. This behavior is explained by the interplay between intense optical fields, large photo-doping, and a small density of states at the K-point with considerable optical phonon energies that block the relaxation into low energetic states. Further, flat bands at the VHS suppress carrier-carrier interactions, and carrier relaxation through coupling with optical phonons becomes the dominant relaxation channel. Consequently, under extreme optical excitation, carrier relaxation is nearly energy independent, leading to a predominant 2D electronic carrier system with a suppressed interlayer hopping[6,57]. The different relaxation dynamics of the strongly-driven non-equilibrium state are reflected in the increase of the optical conductivity as the strong electron-phonon coupling leads to a coherent excitation of optical phonons, which drives the carrier system into a highly conductive state[2,4,5,38]. The here observed transition of graphite into a highly conductive state at room temperature through photo-doping highlights the importance of the electronic occupation near flat bands.

Further, it enhances the understanding of phase transitions in graphite, which are reminiscent of artificially induced superconductivity in the context of twistronics[7,55]. We show that attosecond soft X-ray core-level spectroscopy provides unprecedented insight into the dynamic interplay of carrier types inside materials to elucidate many-body dynamics, quasi-particle interactions, and non-equilibrium physics. Such a direct view of the real-time interplay of many-body interactions will aid in developing optically controlled non-equilibrium states such as light-induced superconductors or optically correlated materials with switchable quantum phases and massively entangled states of light[62].

## Methods

### Experimental setup

A sketch and explanation of the setup can be found in ref. 63 and Supplementary Note 1. Here, we followed the same data acquisition protocol as described in ref. 38.

The data acquisition protocol consists of a series of spectra with different combinations of pump and probe beams on the sample and enables near-shot-noise limited spectra[64]. Shutters along the probe and the pump path are synchronized with the CCD camera and were used to alternate in a sequence of 15 spectra of 40 seconds integration for the pump plus probe ($\tau_{pp}$), 15 spectra of 40 seconds integration for the probe ($\tau_0$) and 4 spectra of 40 seconds for the pump ($\tau_p$) case. This scheme minimizes the influence of possible soft X-ray spectral fluctuations on the analysis, while the pump-only acquisition enables the correct background subtraction. Comparing the probe-only spectra

for a constant delay, no sample heating effects could be observed in the spectra. Each delay time step between the pump and probe resulted in 25 min of measurement time.

We recorded the raw CCD image for each acquisition in a predefined region of interest (ROI), which was identical for $\tau_{pp}$, $\tau_0$, and $\tau_p$. First, all ROIs for a delay step are summed, and then each ROI is added together to obtain the 1D spectrum for each delay step sequence $\tau_{pp}$, $\tau_0$, and $\tau_p$. The detector's dark and thermal noise are removed from each spectrum. To remove the possible residual pump background, the 1D $\tau_p$ spectrum is multiplied by 15/4 to account for the reduced integration time, then subtracted from $\tau_{pp}$ spectrum. To account for possible slow drifts in the soft X-ray flux, despite the 40-second integration windows, we normalize to the energy range between 200-280 eV before the carbon K-edge for $\tau_{pp}$, $\tau_0$, and $\tau_p$, and we apply a 3-pixel boxcar average. These traces are taken to calculate the differential transmission ($\Delta T = T_{pp} - T_0$) normalized to the probe only spectrum, $\Delta T/T_0$.

From the differential transmission spectrum, we can obtain the change of absorption through $\Delta A = -\ln(\frac{\Delta T}{T_0} + 1)$. We used following identity $\frac{\Delta T}{T_0} = \frac{I_0 e^{-\alpha_{pp} d} - I_0 e^{-\alpha_0 d}}{I_0 e^{-\alpha_0 d}} = e^{-\Delta \alpha d} - 1$, with the attenuation coefficient $\alpha$ and sample thickness $d$.

### Data fitting

The time- and energy-dependent absorption spectrum, $A(E, t)$, is described by the product of the calculated density of states, $DOS(E, t)$, and a Fermi-Dirac distribution, $FD(E, T(t))$. Changes to the DOS are accounted for by a shift, $\delta$, and stretching of the energy axis by a. Finally, the fitted spectrum is convoluted with a Voigt function, $\sigma(E, t)$, to account for the detector resolution through a Gaussian broadening of at least 250 meV, and the core-hole lifetime through a Lorentzian with width of 150 meV,

$$A(E, t) = \sigma(E, t) * \left[ 1 - FD(E - E_F(t), T(t)) \right] DOS(a(t)(E - \delta(E, t)))$$

We separately apply this function to the unpumped and pumped spectra and minimize the difference between the fit and measured values at each recorded energy channel using a least-square method. More details can be found in the supplementary information.

## Data availability

Data that support the findings of this study are available from the corresponding authors upon request. Source data are provided as a Source Data file.

## Code availability

The code used for fitting the data is available from the corresponding author upon request.

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

## Acknowledgements

J.B. acknowledges financial support from the European Research Council for ERC Advanced Grant "TRANSFORMER" (788218), ERC Proof of Concept Grant "miniX" (840010), FET-OPEN "PETACom" (829153), FET-OPEN "OPTOlogic" (899794), FET-OPEN "TwistedNano" (101046424), Laserlab-Europe (871124), Marie Skłodowska-Curie ITN "smart-X" (860553), MINECO for Plan Nacional PID2020–112664 GB-I00; AGAUR for 2017 SGR 1639, MINECO for "Severo Ochoa" (CEX2019-000910-S), Fundació Cellex Barcelona, the CERCA Program/Generalitat de Catalunya, and the Alexander von Humboldt Foundation for the Friedrich Wilhelm Bessel Prize. A. S. acknowledges Marie Skłodowska-Curie Grant Agreement No. 754510 (PROBIST). We thank T. Danz and C. Ropers for providing the sample. We would like to thank R. Ernstorfer, B. Rethfeld, and M. Ivanov for their discussions on the subject and J. Menino and C. Dengra for their technical support.

## Author contributions

J.B. conceived and supervised the project. N.D.P., T.P.H.S. and D.E.R. performed the experiments with support from S.S., M.R. and J.B. T.P.H.S. developed the analytical model and analyzed the data. A.S. wrote the code to analyse optical excitation pathways. T.P.H.S. and J.B. wrote the manuscript with contributions from N.D.P.

## Competing interests

The authors declare no competing interests.
