## [Peer Review File · Nature Communications]

REVIEWER COMMENTS

Reviewer #1 (Remarks to the Author):

This manuscript reports the ultrafast carrier dynamics of graphite under strong optical irradiation, probed by attosecond soft X-ray absorption spectroscopy at the carbon K edge. The use of soft X-ray absorption spectroscopy not only allows the authors to disentangle the dynamics of electrons and holes, it also enables access to the temporal evolution of carrier temperatures as well as shifts in the Fermi energy and density of states. With strong driving of graphite with 0.7-eV light, the measurements reveal a peculiar behavior of the relaxation dynamics of electrons, which is attributed to the presence of electronic flat bands. Accessing this feature of the conduction band by intense photoexcitation is also found to enhance the optical conductivity of graphite by one order of magnitude. The work applies a state-of-the-art experimental technique – attosecond soft X-ray absorption spectroscopy – to a contemporary research field – strongly-driven, nonequilibrium condensed matter. The interpretation and conclusions are well-supported by experimental data, and the manuscript is well-written. For these reasons, I recommend publication of a suitably revised manuscript in Nature Communications after the authors address the following comments.

1. Have the authors performed a fluence-dependence study to determine the photon orders involved in the photoexcitation of graphite at both the lowest and highest fluences employed in the experiments?
2. How did the authors arrive at the photon density of $5 \times 10^{22} \text{ cm}^{-3}$? (p. 4, line 130)
3. The authors found that the carrier relaxation times increase with excitation fluence. To what extent is the increase in relaxation time due to the hot phonon effect, without the need to invoke the existence of flat bands?
4. On p. 6, line 182, the authors write that "... carrier-carrier scattering because the main dephasing channel." The use of "dephasing" here is potentially confusing because I believe that the authors are referring to energy relaxation of carriers, not the loss of phase coherence between carriers (?).
5. On p. 6, line 195, the authors write that "this behavior now arises due to interlayer hopping." The manner in which the authors arrive at this statement is unclear. More elaboration would be helpful.
6. Table S1 shows the energy-scaling for different sample dimensionality and different energy dispersions (parabolic vs. linear). However, the table does not show what happens for dispersion-less flat bands.
7. Figure 3: The labels for the panels are missing. I am guessing that they correspond to different excitation fluences, analogous to Figure 2.
8. Under Section S4 of the Supplementary Information, the authors might consider showing some examples of the spectral fits to the equation for $A(E,t)$ that appears in the section. In addition, I am unsure as to whether Fig. S5a is for the electron temperature or the hole temperature? I would

imagine that one should be able to extract separate temperatures for the electrons and holes from the experiments, depending on whether one is probing above or below the Fermi level.

Reviewer #2 (Remarks to the Author):

The manuscript “Enhanced optical conductivity and many-body effects in a semi-metallic light-matter hybrid” by T.P.H. Sidiropoulos et al. presents an experimental investigation of the charge dynamics when a solid state system is driven by a strong NIR laser pulse. This has been achieved by means of attosecond soft X-ray core-level spectroscopy in an NIR/soft-x-Ray pump probe configuration. By investigating the non-equilibrium many-body dynamics of a semi-metallic light-matter hybrid under extreme photo-doping they found conditions at which the optical conductivity can be significantly increased. Additionally they emphasize on the importance of the method towards investigations of many-body dynamics, non-equilibrium physics, aiming to develop optically controlled light-matter hybrids such as light-induced superconductors or optically correlated materials with switchable quantum phases and massively entangled states of light.

The method and the results look very interesting and in principle can justify a publication in Nature Communications. However the manuscript is not well written and the data analysis (especially the error analysis) is not sufficient. Because of this a reader may have many doubts about the validity of the drawn conclusions. For this reason a revised version addressing the following issues is required before I take a decision.

Comments:

1) The manuscript is written in a so complicated way that even an expert in the field is difficult to follow. The authors should provide definitions in the most of the terminologies that they have used and avoid continuously using abbreviations. Also, they have to explicitly state in the Figure captions, which from the data have been obtained by theoretical calculations (and which equations has been used) and which by the experiment.

2) In Fig. 1b the values on the x-axis (absorbance) are missing.

3) In the caption of Fig.1b the authors write "Open symbols are the measured XAFS with (red squares) and without (black circles) NIR excitation and the corresponding fits (dashed lines)". At which delay values between the NIR and the soft-x-Ray pulse these curves have been recorded?

4) In Fig. 1d: I don't understand how these curves have been obtained.

a) In the main text is written that these are line outs of the absorption measurement shown in Fig.1c. At which photon energy values these line outs are taken? This should be explicitly written in the caption of Fig.1d.

b) Considering the values (from -0.2 to +0.2) shown in color bar of Fig.1c, i dont understand how in Fig.1d the values in y-axis are ranging from -0.6 to +0.6, which are much larger than than +-0.2.

c) In Fig. 1c, in the energy range from E_f to 286eV, the values of ΔA are negative up to 0.7 ps and then are getting positive. Why this behavior is not reflected in Fig.1d?

5) At the beginning of page 4 (lines 117-119) the authors write "We infer the sample's intrinsic n-doping and Fermi energy at 285.1 eV from a fit of the absorption spectrum to a room temperature Fermi-Dirac distribution multiplied by the DOS; see SI for details.". Which are the parameters that have been used for the fit? How the error of the measurement influences the time scales obtained by the fit?

6) At the beginning of page 4 (lines 132-134) the authors write "Figure 1d shows lineouts of the absorption measurement under such conditions, exhibiting asymmetric dynamic evolution of the states above (electrons) and below (holes) the Fermi energy.".

I'm not totally convinced about this. The error analysis has not been sufficiently described in the manuscript and thus i cannot judge the validity of this claim.

The authors have to extensively analyze and explicitly discuss in the SI of the manuscript the error analysis. This is absolutely necessary for the justification of the drawn conclusions and frankly speaking this is my main concern about this work. I'm saying this because from the plots of Figs.1c,d, if someone take in to account the fluctuation of raw data, all decay times look the same.

The same is my concern about the data presented in Figs 2,3. For example i'm surprised from the small errors of the data points along the E-Ef axis.

8) In the last sentence of the conclusions the authors write "Such a direct view ... and massively entangled states of light."

The published works that are relevant to this matter are: 1) Nature Phys. 10, 1104–1108 (2021) 2) PRA Phys. Rev. A 105, 033 714 (2022), 3) PRL 128, 123603 (2022) and 4) PRX Quantum 4, 010201 (2023).

Do the authors refer to the works? If yes, they should cite these papers.

Reviewer #3 (Remarks to the Author):

The authors investigate the ultrafast carrier dynamics of highly pyrolytic graphite (HOPG) based on attosecond soft X-ray core-level spectroscopy (XAFS).

They use a generalization of the extended Drude model (EDM) to analyse transient optical data, extract relaxation times, and infer qualitative changes in the opto-electronic properties of HOPG upon absorption of high-fluence driving fields.

The manuscript is very nicely written and it presents a detailed analysis of the transient optical properties of HOPG at unprecedented conditions of optical excitation. Key findings of this work are the observation of an order-of-magnitude enhancement of the optical conductivity as a consequence of photocarrier excitation in the flat bands of HOPG, and the suppression of interlayer hopping at strong driving fields.

Additionally, the authors report a transition between different dimensionalities of the carrier dynamics (involving a suppression of interlayer

carrier hopping at high fluence), which are energy dependent and are influenced by the excitation conditions. These are novel results of broad significance for the field of ultrafast spectroscopy. The methodological aspects are illustrated/discussed in sufficient detail to enable reproducibility and to suggest the absence of major flaws in the data acquisition.

A major issue with this manuscript, however, is the analysis of XAFS data and, in particular, the identification of transient intensities with the relaxation time. In fact, some of the identities used to relate transient optical intensities to the relaxation rates seem to lack a rigorous justification, despite being a central tool for the data analysis throughout this work.

Based on these points, I believe that revisions are required before publication.

I list below a few additional comments.

- In several instances, the authors refer to the light-induced changes of the material as a "light-matter hybrid" (title included). However, there is no evidence of polaritonic states or quantum fluctuations of the electro-magnetic field that may demonstrate the hybridization between matter and light. In other words, there is no hybridization between matter and light and, in fact, all (light-induced) properties are analyzed and explained exclusively by accounting for matter degrees of freedom. Therefore, the system in consideration - despite the extreme excitation fluences considered here - does not seem to give rise to a "light-matter hybrid", but rather a strongly-driven excited state.

- According to the authors "the extreme optical doping increases the relaxation

times by one order of magnitude compared with the single-particle scattering times from photoemission measurements on graphite." This statement, however, does not seem to be corroborated by the data set.

The decay time of the photoexcited electron and hole population, reported in Fig. 1d, changes from 106 to 236 fs between the two values of fluence for electrons, and similarly for holes. Shouldn't the carrier population dynamics reflect changes in the scattering time? If this is the case, how are these differences justified?

- A major issue that I see with this manuscript is the use of equations (1) and (2) as a main tool for analyzing the dataset. In Equation (1), it is true that the scattering time relates to the imaginary part of the self-energy ($\text{Im } S$), however, the second part of the equality is not correct, and the spectral function is not related by a simple proportionality relation to $\text{Im } S$. The same applies to Eq. (2). Because the main results and conclusions of the manuscript rely on use of Eqs. (1-2), more care should be used in establishing a relation between spectral intensities and relaxation times.

- The transition between the dimensionality of the carrier dynamics is based on the comparison between the thermalization rates and the scaling behaviour of the DOS for simple models of solids with parabolic or linearly dispersive bands. The exponent that characterizes the relation between the DOS to the energy (relative to E_F), however, depends strongly not only on the system dimensionality, but also on the details of the bands structure. The band structure of HOPG is more complex than the simple models reported in Table S1, and this is expected to influence strongly the scaling behaviour of the thermalization rate.

As such, it seems unlikely that the different exponents (shown Fig. 2) can give indication of a change in dimensionality of the carrier dynamics. Most likely, this simply reflects a change in DOS due to the more complex bands of HOPG.

We would like to thank the reviewers for their careful consideration of our work and appreciate their valuable feedback. We have carefully considered their comments and address them in a detailed point-by-point response below.

Reviewer #1:

We would like to express our gratitude for the reviewer's thorough assessment of our work and the valuable feedback. We sincerely appreciate the positive remarks and recognition of our findings. We have taken into account the reviewer's suggestions for improvement and made respective changes in the manuscript. We provide our detailed responses below.

R1: Have the authors performed a fluence-dependence study to determine the photon orders involved in the photoexcitation of graphite at both the lowest and highest fluences employed in the experiments?

The reviewer raises an interesting point. We did measure at various fluences, but we have not tried to deduce photon order dependencies due to the strong dipole transition matrix elements and the very efficient carrier scattering mechanisms in graphite. See also our previous work [Phys. Rev. X. 11, 041060 (2021)].

Nevertheless, we can try to address the reviewer's question based on the similarity to multilayer-graphene. In graphene and multilayer graphene, two-photon absorption (2PA) dominates over saturable absorption in the near-IR for intensities exceeding $\sim 2\text{GW}/\text{cm}^2$ [APL 114, 091111 (2019), Opt. Exp. 24 (12), 13033 (2016)]. This intensity is already exceeded for the low fluence case. However, we do not feel confident in making any robust statement for the saturation intensity of 2PA or the onset of 3PA.

Here, we identify the possible photoexcitation pathways by calculating the required photon numbers for a transition into free electronic states considering the calculated electronic band structure of graphite and the measured pump pulse spectrum as discussed in the Supplementary Figure S3. We have taken into account the intrinsic n-doping of our sample, which is crucial, as it limits one photon excitations to states near the K-point. Only through two- and three-photon absorption processes electronic states near the M-point can be directly excited.

We would like to emphasize that multiphoton absorption is only one possible mechanism for the excitation of carriers into states near the M-point. In our previous work [Ref. 38 or Phys. Rev. X. 11, 041060 (2021)], we have shown that carrier multiplication effects, which occur on early timescales, play also an important role in the broadening of the carrier occupation. Although this work focusses on the relaxation of carriers, we discussed the multiphoton excitations to complete the picture of possible excitation mechanisms.

We have added the two references to the manuscript.

R1: How did the authors arrive at the photon density of $5 \times 10^{22} \text{ cm}^{-3}$? (p. 4, line 130)

We apologise for not sufficiently describing how we arrive at the quoted number. This value is a rough estimate of the number of optically excited carriers assuming linear absorption. Here, we followed the same approach as in our previous work [Ref. 38] and calculate the absorbed optically excited carrier number, n_{opt} , taking into account only the linear absorption, α , over the sample thickness, d .

$$n_{opt} = \frac{\alpha I}{\hbar\omega d},$$

with graphite's absorption $\alpha = 0.31$ at a photon energy of $\hbar\omega = 0.7$ eV and an angle-corrected thickness $d = 122$ nm. We would like to point out that the stated fluences reported in this work take the linear absorption of graphite into account. Therefore, we have changed the notation of absorbed photon numbers to be in line with our previous work and added the corresponding citation.

"Such extreme photo-doping with $n_{opt} \approx 5 \times 10^{22}$ electrons/cm³ [Ref. 38] significantly exceeds the number of free electronic states ($\sim 10^{19}$ cm⁻³)¹⁹, thus effectively altering the single-particle electronic structure in highly photo-excited graphite with non-equilibrium carrier dynamics^{27,28}"

We also added a table to the supplement to highlight the difference between the fluence on the sample and the absorbed fluence. Note that in the original manuscript we accidentally stated the incident fluence for the low fluence case instead of the absorbed fluence. This has been corrected.

R1: The authors found that the carrier relaxation times increase with excitation fluence. To what extent is the increase in relaxation time due to the hot phonon effect, without the need to invoke the existence of flat bands?

The reviewer raises an interesting point, and we realize that we should have discussed this aspect in more detail in the manuscript. We had only briefly discussed the relaxation bottlenecks in graphite in the Supplement Material without explicitly mentioning the hot phonon bottleneck.

The main relaxation channels for optically excited carriers in graphite are either through carrier-carrier or carrier-phonon scattering. In graphene and graphite there exist several bottlenecks that slow down carrier relaxation; the large optical phonon energies, the slow relaxation of optical phonons, the low density of states at the K-point, and in our case also the flat bands near the M-point. Which bottleneck then limits carrier relaxation depends on the photon energy, the carrier temperature, and the carrier density [Ref. 49, J.Phys.: Condens.Matter 25, 054201 (2013)], but also the position of the Fermi-energy. For the here used photon energies of 0.7 eV and all three excitation fluences, the hot phonon bottleneck should be dominant. Carriers rapidly thermalize through carrier-carrier scattering and the strong carrier-phonon coupling leads to a rapid increase in the phonon temperature such that both temperatures become similar on ultrashort time scales, as for example shown in our previous work [Ref. 38].

Interestingly, for the low fluence case, the relaxation rate is faster than the expected 200 – 250 fs for these excitation parameters [Refs. 24, 33, 35; Phys. Rev. B 92, 184303 (2015)]. In contrast to the previously reported carrier dynamics in undoped graphite/graphene, our sample exhibits a strong n-doping, thus more states are available near the Fermi-energy. For the low fluence case, this doping initially avoids the bottleneck at the K-point and only becomes important when increasing the fluence to 20 mJ/cm². From the energy integrated relaxation dynamics, as shown in Figure 1d, one would come to the conclusion that indeed the hot phonon bottleneck is the main reason for the decrease in the relaxation rates when increasing fluence. We note that for fluences exceeding 1 mJ/cm² the phonon relaxation time does not increase significantly [Phys. Rev. B 92, 184303 (2015)]. However, from the time-resolved absorption spectrum it becomes clear that in the high fluence case carriers occupy states near the M-point where the bands are flat. Carriers in the flat-bands require a large crystal momentum to lose significant energy for the relaxation to lower states, which is only achievable through carrier-phonon scattering [Ref. 35]. The additional bottleneck induced by the flat-bands is also highlighted in Fig. 2c of the manuscript, where the relaxation rate for carriers in the flat

bands is smaller than the for carriers occupying states below. In fact, without the excitation of carriers into flat-bands the relaxation rate should decrease continuously with decreasing carrier energy, for example see Ref. [24].

Based on the reviewer's comment, we have added the following discussion of the relaxation times from Fig. 1d to the manuscript:

“An exponential fit to the energy-integrated absorption spectrum over the apparent signal, $\Sigma_i \Delta A(E_i, t)$, for three different pump fluences of 3.2 ± 0.2 , 22.8 ± 1.4 and 81.4 ± 5 mJ/cm² reveals that extreme optical doping increases the relaxation times with electrons losing energy faster than holes. Compared to single-particle scattering times from photoemission measurements on graphite, these relaxation times are by one order of magnitude larger and highlight XAFS as a probe of graphite's non-equilibrium many-body response^{18,30,52}. The initial non-thermal energy distribution of optically excited carriers rapidly thermalizes through carrier-carrier scattering and strong coupling to optical phonons³⁸. However, further relaxation of hot carriers slows down due the small density of states at the K(H)-point, the considerable optical phonon energies, and their slow cooling through phonon-phonon scattering with typical reported carrier relaxation times of 200-250 fs [Ref. 24, 33, 35, J.Phys.: Condens.Matter 25, 054201 (2013), Phys. Rev. B 92, 184303 (2015)]. The large intrinsic doping of our sample, however, initially avoids these bottlenecks as the Fermi-energy is above the K-point where the density of states is larger. This is apparent from the short electron (hole) decay time of 111 ± 63 fs (231 ± 67 fs) for the low fluence case. Only by increasing the fluence to 22.8 ± 1.4 mJ/cm² the electron (hole) relaxation time increases to 223 ± 59 fs (309 ± 22 fs) as low energetic states become occupied and carrier relaxation becomes limited by the slow cooling of optical phonons. With a further increase in fluence to 81.4 ± 5 mJ/cm² another bottleneck occurs as carriers occupy flat-bands, where carrier relaxation can only occur through the transfer of a large crystal momentum by scattering with optical phonons [Ref. 35].”

We also extended the discussion in the Supplement:

“In graphene the slow relaxation of photo-doped carriers is a consequence of the low density of states near the K-point, the large optical phonon energies, and the slow cooling of optical phonons through phonon-phonon scattering [S7,S8, 38, 49]. At high fluences in the mJ/cm² regime ultrafast carrier-carrier scattering establishes a hot carrier distribution with a temperature, which, in our case, is close to the photon energy. Band filling due to the low density of states near the K(H)-point and the large optical phonon energies limit the relaxation of low energetic carriers. [Ref. 49, J.Phys.: Condens.Matter 25, 054201 (2013)]. For carriers with energies larger than the optical phonon energy, relaxation through the strong coupling of electrons to optical phonons becomes possible, rapidly increasing phonon temperatures. Consequently, the slow cooling of the hot optical phonon distribution limits carrier relaxation [Ref. 38, Phys. Rev. B 92, 184303 (2015)] For the high fluence case, another mechanism has to be considered as carriers are excited into states near the VHS where the density of states is large and bands become flat. The flat-band states require carriers to transfer a large momentum in order to significantly lose energy. Thus, carrier thermalisation through carrier-carrier scattering is suppressed and scattering with strongly coupled optical phonons becomes the main relaxation channel.”

R1: On p. 6, line 182, the authors write that “... carrier-carrier scattering because the main dephasing channel.” The use of “dephasing” here is potentially confusing because I believe that the authors are referring to energy relaxation of carriers, not the loss of phase coherence between carriers (?).

We agree with the reviewer that the term dephasing in the context of this work may be confusing. Here, again we were referring to our previous work [Ref. 38], where light-field-driven carrier oscillations during the excitation with the pump pulse and subsequent dephasing were discussed. In this work only later timescales are discussed. We therefore, replaced dephasing with relaxation. The sentence now reads:

“In contrast, for holes, the split-off bands near the K-point, and the linear bands between the K-point and the hole VHS, lead to vastly different relaxation rates as carrier-carrier scattering becomes the main relaxation channel.”

R1: On p. 6, line 195, the authors write that “this behavior now arises due to interlayer hopping.” The manner in which the authors arrive at this statement is unclear. More elaboration would be helpful.

We apologize for our brevity in explaining this statement. It is based on our discussion at the beginning of the manuscript where we describe the band structure of graphite:

“In both systems, interlayer coupling of electronic states in neighbouring layers lifts the degeneracy of the bands at the Dirac point, which leads to split-off bands with a band-gap of less than 60 meV at the K-point^{18,19}. In contrast, the bands remain linear around the H-point, i.e., charge carriers behave like massless Dirac-particles²⁰. The split-off bands at the K-point and the H-point are connected by a near dispersion-less band, removing the electronic system's two-dimensional (2D) confinement; thus, through interlayer coupling the carriers occupy 3D Fermi surfaces^{19,21–23}.”

We like to elaborate on the effect of interlayer coupling on the dimensionality of the carrier system:

In AB stacked graphite the interlayer coupling between graphene sheets of $\gamma \approx 0.3 - 0.4$ eV lifts the degeneracy of the bands at the K-point with an offset for the split-off bands of around $\pm 2\gamma$ [Ref. 53, Ref. 55]. As the carrier dispersion at the K-point is connected with the H-point, the partial opening of a gap at the K-point, consequently leads to a charge carrier dispersion along the k_z -direction and the Fermi surfaces become 3D [Refs. 21, 22, 53]. However, as the Dirac point at the K-point lies energetically below the H-point the Fermi surfaces become nested with connecting points along K-H [Refs. 19, 21, 22, Schneider PRL 108, 117401 (2012)]. Through these (band touching) points, electron and hole pockets emerge at the K- and H-points in undoped graphite, respectively. Thus, the interlayer interaction in graphite leads to 3D Fermi surfaces, and only for carrier states near the K(H)-point a 2D-like character, as known from graphene, is preserved [Refs. 19, 20, 21].

We like to emphasize that with XAFS we probe the momentum-integrated response of a highly photo-excited, non-equilibrium many-body system. Thus, the scaling of the relaxation dynamics presented here, does not reflect the individual carrier dispersion in the different bands at the K- and H-point but rather the overall behaviour of the many-body system. This is similar to most other metals, where no clear feature of the electronic bands on the carrier relaxation rate with energy is observed; these typically show the quadratic energy scaling as expected from a 3D Fermi-liquid [Phys. Rev. B 61, 13 484 (2000)].

Now, for our sample, the Fermi-energy is close to the bottom of the split-off band, which is parabolic. The observed $E^{1/2}$ -scaling of the relaxation rate highlights the 3D character of the carriers through the connection of the split-off bands with the H-point.

For energies close to the K/H-point, the nesting of the Fermi surfaces creates carrier pockets where the carriers are confined to 2D. This is supported by the linear scaling of the relaxation rate.

In the medium and high fluence measurements we observe a change from a linear to a quadratic scaling of the relaxation rate for holes with energies of about 0.7 eV below the K-point which is close to 2γ . Thus, for hole energies exceeding this value, interlayer hopping becomes possible; this was reported by Ohta et al. [Ref. 53], who observed a modulation of the out-of-plane momentum with a periodicity of the interlayer spacing for carriers with an energy of 1 eV. Again, the interlayer coupling opens up an out-of-plane carrier dispersion, connecting the split-off bands at the K-point with the linear bands at the H-point, and the Fermi surface is again 3D. The observed quadratic scaling in this energy ranges highlights the linear bands.

In order to further clarify our statement, we have edited our discussion regarding the scaling of the relaxation rate:

“Finally, we leverage the similarity between the self-energy and the single particle density of states [Ref. 43] to assess the general charge carrier dispersion over the apparent energy range and the dimensionality of the charge carrier system from the scaling of the relaxation rates (see Tab. 3 in the SI). We find that near the Fermi energy, where the split-off bands approximate parabolic behaviour the $E^{1/2}$ scaling of $1/\tau(E)$ indicates a 3D hole system. This pinpoints the 3D Fermi surface to the H-point. We note that this finding agrees with existing magneto-transport and photoemission measurements²⁰⁻²². Further from the Fermi energy, around the K(H)-point, we infer from the linear scaling a change in the dimensionality to a 2D-like carrier system with linear dispersion³³. However, specifically for energies approaching the VHS, holes behave again like a 3D system, as carriers occupy states beyond the split-off bands, arising through interlayer coupling, and where the bands approaching the VHS become linear⁵³”.

R1: Table S1 shows the energy-scaling for different sample dimensionality and different energy dispersions (parabolic vs. linear). However, the table does not show what happens for dispersion-less flat bands.

The reviewer raises an interesting point. The derivation of the scaling of the density of states for different dispersion and dimensionality, shown in the Supplementary Material, is based on the volume of electronic states in k-space and the carrier dispersion. The density of states is a function of the inverse group velocity of the charge carriers and thus diverges for a vanishing group velocity. For perfectly flat bands, the group velocity does become zero and the density of states is then described by a Dirac-delta function in n-dimensions.

However, as van Hove already pointed out in his seminal paper on the phonon dispersion in crystals, the exact scaling depends on the type of flat band, that is a saddle point or another type of extrema as defined by the second derivative of the carrier dispersion [Phys. Rev. 89 (6), 1189 (1953)]. For a saddle point the density of states then has a logarithmic divergence, a so-called ordinary van Hove singularity. With the advent of topological insulators, the topic of flat bands became topical and today other higher order van Hove singularities are discussed, each resulting in a different scaling of the density of states [Kulynych PRB 106, 045115 (2022), Yuan, Nat. Comm. 10, 5769 (2019)]. Graphite exhibits two different types of van Hove singularities in the relevant energy range, one extremum at the K-point and a saddle point at the M-point. This is the reason why we refrained from stating a general term for the density of states near flat bands since this would be misleading the reader. Based on the question of the reviewer, we extended the discussion on the scaling of the density of states to include the ambiguity of the van Hove singularity:

*“For a carrier dispersion with a vanishing slope ($p = 0$), a classical derivation of the DOS would yield a Dirac-Delta function in n -dimensions. However, the exact scaling of the DOS depends on the type of flat-band as defined by the curvature of the carrier dispersion. In graphite, the carrier dispersion vanishes at the extrema near the K -point and near the M -point, where no simple scaling laws for the density of states are possible. We refer to the literature for a detailed discussion on the scaling of the density of states near extrema in the carrier dispersion in graphite [Ref. 28, *Bena Phys. Rev. B* 72, 125432 (2005)] and for flat bands in general [Kulynych *PRB* 106, 045115 (2022), Yuan, *Nat. Comm.* 10, 5769 (2019).]”*

R1: Figure 3: The labels for the panels are missing. I am guessing that they correspond to different excitation fluences, analogous to Figure 2.

We thank the reviewer for pointing out the missing labels for the different fluences. We have added these to Fig. 3.

R1: Under Section S4 of the Supplementary Information, the authors might consider showing some examples of the spectral fits to the equation for $A(E,t)$ that appears in the section. In addition, I am unsure as to whether Fig. S5a is for the electron temperature or the hole temperature? I would imagine that one should be able to extract separate temperatures for the electrons and holes from the experiments, depending on whether one is probing above or below the Fermi level.

We thank the reviewer for this suggestion and we have added examples of the fits to the Supplement. We would also like to point the reviewer's attention to the Supplement Videos showing the fits to the experimental data for each pump-probe delay.

The fit to the x-ray absorption spectra only takes one Fermi energy for the whole charge carrier system into account. We agree that a model that treats electron and hole temperatures separately would be interesting. However, graphite is a semi-metal and it becomes metallic under strong optical excitation. Furthermore, the Fermi energy in our sample lies in the bands above the K -point. Thus, a clear separation into electrons and holes, as known from semiconductors, might not be appropriate. In undoped graphite, however, such a separation has been reported before [Ref. 24, *Phys. Rev. B* 92, 184303 (2015)]. In doped graphite, initially after optical excitation also a non-thermal electron and hole distribution exist that rapidly thermalize due to the fast carrier-carrier scattering time of graphite. The initial non-thermal carrier distributions spread in energy on ultrashort timescales and, together with a small photon energy of 0.7 eV, it is not possible to separate the distribution with the limited spectral resolution of our soft x-ray spectrograph. From the videos in the Supplement at early time-scales a separate electron-hole distribution might be apparent as a broad peak below the Fermi-energy, similar to previously observed features on undoped graphite using ARPES see [Ref. 24, *Nat. Mat.* 12, 1119 (2013)], however these features are not clear enough to allow reasonable fitting with two independent Fermi-Dirac distributions.

Reviewer #2

We very much appreciate the reviewer's positive assessment of our work. We have addressed the reviewer's concerns and hope that our response will answer the reviewer's questions.

R2: The manuscript is written in a so complicated way that even an expert in the field is difficult to follow. The authors should provide definitions in the most of the terminologies that they have used and avoid continuously using abbreviations. Also, they have to explicitly state in the Figure captions, which from the data have been obtained by theoretical calculations (and which equations has been used) and which by the experiment.

We apologize for the possible confusion due to trying to describe the intricate interplay in a compact way. We understand that using many abbreviations and possible lack of additional descriptions may have aggravated the notion. As response, we thoroughly went through the manuscript and removed abbreviations were possible. This includes removing all abbreviations for soft x-rays and near-infrared. We reduced the usage of DOS as an abbreviation for the density of states. Where possible, we replaced HOPG with graphite, and VHS with flat bands.

To remove the confusion of what is experimental data and what is obtained from a fit we have reworked the descriptions. In general, experimental data are symbols and data from calculations are continuous lines. We went through all figure captions, and further clarified the origin of the presented data.

Figure 1 caption *"a, Single-particle electronic band structure of AB stacked highly pyrolytic graphite (HOPG) calculated with Wien2K⁴⁸. The energy is referenced to the K-point. The near-infrared beam excites carriers around the Fermi energy (E_F), and the soft X-ray beam probes changes in the carrier occupation by promoting 1s core-electrons that are bound by 284.2eV into free electronic states around E_F (horizontal line). b, Retrieved density of states from the fit for the unpumped case (light blue line) and pumped case, 15fs after near-infrared excitation with $81.4 \pm 5 \text{ mJ/cm}^2$ (light red line). Symbols are the measured XAFS with (red squares) and without (blue circles) near-infrared excitation and the corresponding fits (solid lines). c, ΔA for excitation with $81.4 \pm 5 \text{ mJ/cm}^2$ (high fluence case). The black line indicates the position of the static Fermi energy at 285.1 eV, as retrieved from the fit. The red (blue) line at 287.6 eV (282.2 eV) marks the upper (lower) energy limit for the lineouts in d. d, $\Sigma_i \Delta A(E_i)$ above (negative) and below (positive) the static E_F from c for the three measured fluences and the corresponding decay times from exponential fits (solid lines). The error bars are the relative error of the mean of $\Delta A(E_i)$ in the corresponding electron and hole energy range."*

Figure 2 caption: *"Carrier thermalisation rate $1/\tau_{th}$ from an exponential fit to ΔA for all three measured fluences (symbols) for electrons a-c, holes e-g, and from the fitted absorption spectra (solid lines). The energy axis is referenced to the static Fermi energy. The horizontal lines in e-g indicate the position of the K-point. The grey lines indicate the slope of $1/\tau$ in the apparent energy region (coloured boxes), highlighting the dimensionality of the carrier system. d, h, The renormalized density of states retrieved from the model at the delay time at which the carrier temperature becomes maximal."*

Figure 3 caption: *"a-c, Mass enhancement factor $\lambda(E)$ and optical conductivity $\sigma(E)$ (d-f) retrieved from the relaxation rate, $1/\tau(E)$, using a Kramers-Kronig transformation of the experimental data (symbols) and the model (lines). The vertical line in d-f marks the universal optical conductivity of a single graphene layer at $\pi G_0/4$. The light line in d-f is the Drude conductivity."*

R2: In Fig. 1b the values on the x-axis (absorbance) are missing.

We thank the reviewer for pointing this out. We are now showing the normalised transmitted signal and have added the values to the x-axis.

R2: In the caption of Fig.1b the authors write "Open symbols are the measured XAFS with (red squares) and without (black circles) NIR excitation and the corresponding fits (dashed lines)". At which delay values between the NIR and the soft-x-Ray pulse these curves have been recorded?

We apologize for this confusion and we have added a label to the figure stating the measured delay and edited the figure caption:

"Retrieved density of states from the fit for the unpumped case (light blue line) and pumped case, 15fs after near-infrared excitation with $81.4 \pm 5 \text{ mJ/cm}^2$ (light red line). Symbols are the measured XAFS with (red squares) and without (blue circles) near-infrared excitation and the corresponding fits (solid lines)."

R2: In Fig. 1d: I don't understand how these curves have been obtained.

We would like to thank the reviewer for the input and hope with the implemented changes to panels c) and d) the presented data will be clear now.

R2: a) In the main text is written that these are line outs of the absorption measurement shown in Fig.1c. At which photon energy values these line outs are taken? This should be explicitly written in the caption of Fig.1d.

We understand that this has not been clearly explained. In panel c) we are now highlighting the integration regions for electrons and holes. We also added the corresponding regions to the low and medium fluence scans shown in the Supplement. We have added this energy also to the caption:

"The red (blue) line at 287.6 eV (282.2 eV) highlights the region for the lineouts in d."

R2: b) Considering the values (from -0.2 to +0.2) shown in color bar of Fig.1c, i dont understand how in Fig. 1d the values in y-axis are ranging from -0.6 to +0.6, which are much larger than than +0.2.

We apologise for the confusion due to our mistake in labelling. The figure shows the sum of ΔA in the indicated energy region. We have changed the y-axis label to $\Sigma_i \Delta A(E_i)$ to make the distinction clear. We also believe that the reviewer seems to expect an average value here (which would be in the range of +/- 0.2), thus we would like to emphasize an important aspect of our work:

XAFS at the K-edge is sensitive to the change in carrier occupation, thus a sum over the apparent energy range highlights the optically excited number of carriers [We kindly refer the reviewer to our previous work on carrier multiplication in graphite, Ref. 38]. The excited number of carriers is crucial for understanding the renormalisation of the single particle band structure and the probability of multiphoton absorption. Moreover, the limited number of available states at low energies near the K-point in graphene and graphite is, besides the slow relaxation and large energy of optical phonons, crucial to understand why carrier relaxation slows down with increasing fluence. Hence, carriers eventually get excited into free electronic states near the van Hove singularity at the M-point, where

interesting correlated many-body effects occur. These appear as clear signatures in the optical conductivity.

R2: In Fig. 1c, in the energy range from E_f to 286eV, the values of ΔA are negative up to 0.7 ps and then are getting positive. Why this behavior is not reflected in Fig.1d?

We greatly appreciate the reviewer's question on how the lineouts in Fig. 1d were calculated. The change in sign of the absorbance for states above the Fermi energy is a peculiarity of the high fluence measurement as carriers occupy states beyond the peak in the density of states. Consequently, the large carrier density causes a renormalization of the electronic bandstructure, see Fig. S8 and Fig. S11. Here, we are interested in the total change in absorbance, which is indicative to the optically excited carrier number, see also the response to the previous point. We therefore are showing the positive (negative) absolute change in signal for states below (above) the Fermi energy. We have added a sentence to explain this:

"Note, for an excitation with 81.4 ± 5 mJ/cm² we have considered the total positive (negative) change in ΔA for states below (above) the Fermi energy."

R2: At the beginning of page 4 (lines 117-119) the authors write "We infer the sample's intrinsic n-doping and Fermi energy at 285.1 eV from a fit of the absorption spectrum to a room temperature Fermi-Dirac distribution multiplied by the DOS; see SI for details.". Which are the parameters that have been used for the fit? How the error of the measurement influences the time scales obtained by the fit?

We thank the reviewer for the comment which helped us to make clear how the intrinsic n-doping was obtained. We fit the absorption spectrum of unexcited graphite with the equation for $A(E,t)$ stated in the Supplement, where the Fermi energy is a free parameter. For the static fit other parameters such as the shift of the density of states δ , and the energy scaling α are kept fixed. The broadening parameter σ is 350 meV to account for the Gaussian broadening of the detector (250 meV) and the core-hole lifetime of graphite (150 meV) as stated in the Supplement. We have added a table with the fitting parameters for the unpumped spectra.

Here, for each measured delay step the x-ray absorption spectrum is fitted by the equation Eqn. S1 and the best fit is found through a least-square method to each spectral data point. The error in the spectral fits arise due to spectral features not perfectly captured by the fit, these are for example spectral noise, and a non-thermalised carrier distribution. In order to reduce spectral noise, we record at each time-delay alternating absorption spectra of optically excited and non-excited graphite. With this measurement protocol we reach a near shot-noise limited spectrum [Summers et al., *Ultrafast Science* 3, 0004 (2023)].

The relaxation rate for each energy is then obtained by fitting an exponential function to the spectral fits from each measured time delay. The retrieved relaxation rate is therefore obtained from multiple individual measurements, which reduces the influence of a potential bad delay scan.

We have added error bars to the retrieved fit parameters shown in S7 and to the retrieved thermalisation rates from Fig. 2.

R2: At the beginning of page 4 (lines 132-134) the authors write "Figure 1d shows lineouts of the absorption measurement under such conditions, exhibiting asymmetric dynamic evolution of the

states above (electrons) and below (holes) the Fermi energy.". I'm not totally convinced about this. The error analysis has not been sufficiently described in the manuscript and thus i cannot judge the validity of this claim. The authors have to extensively analyze and explicitly discuss in the SI of the manuscript the error analysis. This is absolutely necessary for the justification of the drawn conclusions and frankly speaking this is my main concern about this work.

I'm saying this because from the plots of Figs.1c,d, if someone take in to account the fluctuation of raw data, all decay times look the same. The same is my concern about the data presented in Figs 2,3. For example i'm surprised from the small errors of the data points along the E-Ef axis.

We understand that the description of our error analysis and the displayed measurement errors were insufficient. We have remedied this by replotting the figures with error bars, where missing. And, we have rewritten the description of the data acquisition and data handling in the manuscript in lieu of only referencing our previous analysis.

We note that all measurements were performed in such a way as to limit the effects of external noise sources and by following the error propagation law to determine errors. Briefly, to acquire experimental data we recorded multiple CCD images alternating between soft x-ray probe only, soft x-ray probe plus near- infrared pump, and near-infrared pump only. From each spectrum we subtracted the detector's dark and thermal noise. For the pumped case, the background from the pump pulse was also subtracted from the spectra. We have chosen the measurement times and sequence of pump-off/on and background after assessing the noise characteristics of our detection setup for an optimal trade-off between shot, readout and intrinsic noise. See also our previous work, Ref. 38, Summers et al. Ultrafast Science 3, 0004 (2023).

We illustrate the outcome of this procedure for the high fluence measurement since it provides the worst case with highest noise. We measure 15 frames for each delay step and we do not simply sum all 15 frames together. Instead, we calculate the mean and we propagate the standard deviation of the mean to obtain an error of the differential absorption for each recorded energy value and delay step. This is shown in the Supplement in Fig. S4b. The error is greatest near the VHS, around 286 eV, where absorption is strongest and thus the recorded transmitted spectrum is minimal.

Figure S4c shows the lineouts from Fig. S4a over the same energy range as in Fig. 2d. We note that the relaxation time obtained by fitting these lineouts matches within the error with the values obtained when calculating absorption changes by summing all recorded frames.

Figure S4 Error estimation for ΔA . **a**, Absorption changes for the high fluence case as shown in Fig. 2c, however using the mean of 15 spectra instead of the sum. **b**, Error of the absorption changes by propagating the standard deviation of the mean of the 15 recorded spectra at each delay step. **c**, Lineouts from the absorption changes in **a** over the same energy range as used in Fig. 1d.

For completeness we add in the supplement information on the fluctuation of the measurement signal for the different pump cases.

“As described in the Method section we record 15 unpumped and pumped spectra in alternating order to reduce fluctuations inherent to the HHG source. In Fig. S3 we show that this procedure reduces intensity fluctuations below the observed absorption changes with no dependence on the pump-probe delay. For each delay we calculate the difference between the total unpumped and pumped counts in the signal region (280 eV – 300 eV) normalized to the mean of both.

Figure S3 Difference between the total unpumped and pumped counts, ΔI , in the signal region between 280 eV – 300 eV, normalized to the mean of both, \bar{I} , for the low fluence case in **a**, the medium fluence in **b**, and the high fluence case in **c**. The black solid line is the mean across all scans and the grey area the standard deviation of the mean.

The reviewer writes that the data in Fig. 1c,d would look the same if the fluctuations were taken into account. We strongly disagree and believe that this statement comes from us overloading the plot with overlapping graphs. We plotted the carrier relaxation for different conditions together to facilitate the comparison, but this overloaded the graph in the way we displayed data.

We have changed the display of data and the aspect ratio of the figure such that the differences between the scans in Fig. 1d become clear. We believe that the differences between lineouts for the three fluences shown in Fig. 1d and their fits are now clearly visible.

For completeness, we also show comparison between the sum and the mean value together with their standard deviation for electron and hole energy ranges for the three fluences. Within the error we obtain similar carriers relaxation dynamics.

We emphasize though that the mean value is not a meaningful parameter since it implies a homogenous energy distribution of optically excited carriers. Since we cannot make such assumption, we purposely show the sum in the manuscript and not the mean.

Comparison of the carrier relaxation dynamics obtained between the sum of ΔA over the electron and hole energy range (left panel) and the mean of ΔA across the same energy range (right panel).

Regarding the error along the energy axis in Fig. 2,3, we have now included those error bars as well. We note that the main error is the instrumental uncertainty of 250 meV, stated in the manuscript.

R2: In the last sentence of the conclusions the authors write "Such a direct view ... and massively entangled states of light."

The published works that are relevant to this matter are: 1) Nature Phys. 10, 1104–1108 (2021) 2) PRA Phys. Rev. A 105, 033 714 (2022), 3) PRL 128, 123603 (2022) and 4) PRX Quantum 4, 010201 (2023). Do the authors refer to the works? If yes, they should cite these papers.

We thank the reviewer for pointing out these references. We have added the latest reference to the manuscript.

Reviewer #3

We are glad to read that the reviewer acknowledges the novelty and significance of our work. We thank the reviewer for the time spent in evaluating our work and for the helpful comments. We address them below:

R3: In several instances, the authors refer to the light-induced changes of the material as a "light-matter hybrid" (title included). However, there is no evidence of polaritonic states or quantum fluctuations of the electro-magnetic field that may demonstrate the hybridization between matter and light. In other words, there is no hybridization between matter and light and, in fact, all (light-induced) properties are analyzed and explained exclusively by accounting for matter degrees of freedom. Therefore, the system in consideration - despite the extreme excitation fluences considered here – does not seem to give rise to a "light-matter hybrid", but rather a strongly-driven excited state.

We used the terminology light-matter hybrid since the system exists in the measured state only due to the strong light-matter interaction with its properties deviating significantly from any single-particle description. To call this state a strongly-driven excited state also does injustice to describe the physics since an excited state is described in a single-particle framework. We do however see the reviewer's point and in order to avoid confusion with coupled light-matter states, we have replaced the term in the manuscript with "strongly-driven photo-excited state"

R3: According the authors "the extreme optical doping increases the relaxation times by one order of magnitude compared with the single-particle scattering times from photoemission measurements on graphite." This statement, however, does not seem to be corroborated by the data set. The decay time of the photoexcited electron and hole population, reported in Fig. 1d, changes from 106 to 236 fs between the two values of fluence for electrons, and similarly for holes. Shouldn't the carrier population dynamics reflect changes in the scattering time? If this is the case, how are these differences justified?

We believe that the point raised by the reviewer is a misunderstanding due to our wording. The reviewer is correct in the interpretation of Fig 1d, but we aimed to compare our measurements to known single-particle scattering times. The sentence cited by the reviewer was to be considered in combination with its succeeding sentence which states: "*These relaxation times highlight the sensitivity of XAFS to the optical conductivity*^{18,30,50}". We simply aimed to state that the relaxation time of the many-body system measured by us with XAFS is one order magnitude larger than the single particle scattering time measured with ARPES.

The scattering time is a property of a single particle state. The thermalisation time, or relaxation time, is a property of a non-equilibrium many-body system, such as the occupation of optically excited carriers. In order for a many-body system to relax, each individual particle has to undergo a relaxation process, such as scattering with other particles. Therefore, it becomes clear that scattering times and relaxation times are not identical and relaxation times should be different (longer). For a more detailed derivation of the differences between scattering and relaxation times we refer to Ref. 50. To clarify our statement, we have changed the sentences to:

An exponential fit to the energy-integrated absorption spectrum over the apparent signal, $\Sigma_i \Delta A(E_i, t)$, for three fluences of $3.2 \pm 0.2 \text{ mJ/cm}^2$, $22.8 \pm 1.4 \text{ mJ/cm}^2$ and $81.4 \pm 5 \text{ mJ/cm}^2$ reveals that extreme

optical doping increases the relaxation times with electrons losing energy faster than holes. Compared to single-particle scattering times from photoemission measurements on graphite, these relaxation times are by one order of magnitude larger and highlight XAFS as a probe of graphite's many-body response^{18,30,50}.

R3: A major issue that I see with this manuscript is the use of equations (1) and (2) as a main tool for analyzing the dataset. In Equation (1), it is true that the scattering time relates to the imaginary part of the self-energy ($\text{Im } S$), however, the second part of the equality is not correct, and the spectral function is not related by a simple proportionality relation to $\text{Im } S$. The same applies to Eq. (2). Because the main results and conclusions of the manuscript rely on use of Eqs. (1-2), more care should be used in establishing a relation between spectral intensities and relaxation times.

We politely disagree with the notion of the reviewer that Eqns. 1 and 2 are the main tool to analyse the datasets. Our intention here was to simply state that the single particle scattering time is a function of $A(k, \omega)$, in contrast to the many-body relaxation time, which should be a function of $\partial_t \Delta A(E, t)$. And we note that while the exact functional is known for the single-particle scattering time in Eq. (1), the exact relation for the memory function is unknown, as stated in Ref. 46. We do agree with the reviewer that $\text{Im} \Sigma$ is not simply proportional to the spectral function and we did not mean to state this.

To avoid confusion, we have removed the proportionality sign from both equations and instead simply state the function for $A(k, \omega)$:

“Considering that the self-energy, $\Sigma(k, \omega)$ describes the polarization of the electronic system and quasi-particle excitations such as carrier-carrier or carrier-phonon interactions^{29–32}, the direct relation between the self-energy and the inverse of the single-particle scattering time connects the measured single-particle spectrum to the quasi-particle dynamics

$$\frac{1}{\tau_{nk}} = -\frac{2\Im\Sigma(E_{nk})}{\hbar},$$

where the spectral function is related to the self-energy through $A(\mathbf{k}, E) = -2\Im\Sigma(E_{nk}) / [(E - E_{nk} - \Re\Sigma(E_{nk}))^2 + \Im\Sigma(E_{nk})^2]$.”

For the case of the memory function where the exact function is not known we now write:

“And similar to the self-energy, quasi-particle excitations are described through the memory function M ^{5,45–47} with

$$\frac{1}{\tau(E)} = \frac{\Im M(E)}{\hbar}.$$

This equation resembles Eqn. (1), and indeed such a relation between $\Im M(E)$ and $\Im\Sigma(E_{nk})$ has been predicted⁴⁶. While the exact relation between the spectral function and the memory function is not known, the many-body relaxation rate can be obtained from the decay of the carrier occupation, $\partial_t \Delta A(E, t)$.”

R3: The transition between the dimensionality of the carrier dynamics is based on the comparison between the thermalization rates and the scaling behaviour of the DOS for simple models of solids with parabolic or linearly dispersive bands. The exponent that characterizes the relation between the DOS to the energy (relative to E_F), however, depends strongly not only on the system dimensionality, but also on the details of the bands structure. The band structure of HOPG is more complex than the simple models reported in Table S1, and this is expected to influence strongly the scaling behaviour of the thermalization rate. As such, it seems unlikely that the different exponents (shown Fig. 2) can give indication of a change in dimensionality of the carrier dynamics. Most likely, this simply reflects a change in DOS due to the more complex bands of HOPG.

We agree with the reviewer that the scaling of the relaxation rate does depend on the dimensionality of the carrier system and on the band structure. We also agree with the reviewer that for a small energy range such a scaling would not be clear and other band structure effects, such as flat bands or band openings will lead to deviation from this scaling. In contrast, over a large energy range, as measured in our manuscript, the overall dimensionality of the carrier system does reflect in the scaling of the relaxation rate, thus scaling is meaningful.

For instance, Refs 33 and 35 refer to the scaling of carrier relaxation rates in graphite with observed deviation from the expected quadratic scaling of a Fermi liquid for electrons. A similar study exists for holes using angle-resolved photoemission spectroscopy [Ref. 18]. For energies close to the Fermi energy, a near-linear scaling was reported, which was attributed to the linear bands, and near the flat-bands at the M-point a constant relaxation rate was observed [Ref. 33, 35]. Thus, the band structure indeed plays an important role in the scaling of the relaxation rate, but more importantly quasi-particle interactions, which are described by $\text{Im}\Sigma$, influence the carrier relaxation dynamics [Ref. 33, 34, 43]. Yet, $\text{Im}\Sigma$ in graphite, scales similar to the DOS, see Ref. 43. However, that along certain crystal directions and for individual quasi-particle interactions $\text{Im}\Sigma$, shows small variations, see Ref. 33 and 34. As the scaling of the DOS itself depends on the type of bands and the dimensionality of the carrier system, this scaling should be preserved in the relaxation dynamics.

To emphasize that the scaling of the relaxation rate is only an indication of the carrier dimensionality over a large energy range, we have edited the discussion and highlighted the apparent energy regions in Fig. 2:

“Finally, we leverage the similarity between the self-energy and the single particle DOS [Ref. 43] to assess the general charge carrier dispersion over the apparent energy range and the dimensionality of the charge carrier system from the scaling of the relaxation rates (see Tab. 3 in the SI). We find that near the Fermi energy, where the split-off bands approximate parabolic behaviour the $E^{1/2}$ scaling of $1/\tau(E)$ indicates a 3D hole system. This pinpoints the 3D Fermi surface to the H-point. We note that this finding agrees with existing magneto-transport and photoemission measurements²⁰⁻²². Further from the Fermi energy, around the K(H)-point, we infer from the linear scaling a change in the dimensionality to a 2D-like carrier system with linear dispersion³³. However, specifically for energies approaching the VHS, holes behave again like a 3D system, as carriers occupy states beyond the split-off bands, arising through interlay coupling, and where the bands approaching the VHS become linear⁵³. “

We disagree with the reviewer that the band structure of HOPG near the K(H)-point is insurmountably complex. It is certainly simple compared with other metals [see for example Phys. Rev. B 61, 13 484

(2000)], see Fig. 1a. However, we agree with the reviewer that the split-off bands and the nested Fermi surfaces around the K(H)-point add complexity to the system. Presently, unravelling these features is unfortunately below the experimentally achievable spectral resolution.

REVIEWERS' COMMENTS

Reviewer #1 (Remarks to the Author):

The authors have made suitable revisions to their manuscript in response to the comments from the reviewers. I recommend publication of the manuscript in its present form.

Reviewer #2 (Remarks to the Author):

The authors have sufficiently answered my comments.

The readability of the manuscript has been significantly improved. The novelty and the impact of the work are visible.

Additionally, the findings are well supported by the experimental data including sufficient error analysis.

I recommend publication in Nature Communications.

Reviewer #3 (Remarks to the Author):

The authors have responded exhaustively all the points raised in my previous report, and they successfully addressed all my concerns. I recommend the manuscript for publication.